



# Reconstruction of ancient drought in Northwest China and societal responses: A case study of 1759

Zhixin Hao[1,2], Haonan Yang[3], Meirun Jiang[3], Danyang Xiong[3], Jingyun Zheng[1,2]

[1] Key Laboratory of Land Surface Pattern and Simulation, Institute of Geographic Sciences and Natural Resources Research, Chinese Academy of Sciences, Beijing 100101, China
[2] University of Chinese Academy of Sciences, Beijing 100049, China
[3] School of Geography and Planning, Ningxia University, Yinchuan 750021, China

*Correspondence to*: Jingyun Zheng (zhengjy@igsnrr.ac.cn)

**Abstract.** According to the written records and scores of grain harvest in the official historical documents of the Qing Dynasty in China, the spatial-temporal distribution of and the impact caused by the 1759 AD drought in Northwest China were reconstructed, and the response of Chinese society to the drought at that time was summarized. In the spring and summer of 1759, vast areas of northern China suffered from drought, of which that experienced in the northwest region was the most serious. Starting from 27 April 1759, droughts covered Zhili, Shanxi, Shaanxi, and Gansu provinces, and the droughts in Gansu and northern Shaanxi provinces lasted until 23 July 1759. This severe drought caused the worst crop failure in Northwest China in 1759 during the period 1730–1900. By implementing a series of disaster relief measures, the Qing government managed to ease the adverse societal impact of the drought in the summer of 1760. Among the relief measures, tax reduction and exemption in disaster-stricken areas, grain storage in the northwest region, and bumper agricultural harvests in Henan, Shandong, Jiangsu, and Anhui provinces in 1759 were the main reasons for the rapid recovery from the drought impacts. With better climatic conditions in the 18th century, China had higher agricultural harvest levels in the 18th century than in the 19th century. Favorable financial conditions increased drought relief efforts, which was the background for the greater societal adaptability to the climate disaster of 1759.

## 1 Introduction

Drought threatens human food security by reducing crop yields. It is a serious natural disaster that threatens human life and can even lead to the overthrow of regimes. During the period 1997–2016, major drought events occurred 89 times in Asia, directly causing thousands of deaths, and the direct economic losses reached more than US$40 billion (Sapir, 2017). The East African drought in the 1970s–1990s resulted in 250,000 deaths and tens of millions of refugees (Haywood et al., 2013). Historically, simultaneous multi-year droughts in Asia, Brazil, and Africa during the period 1875–1878 caused widespread crop failures and triggered famines on a global scale that killed more than 50 million people (Singh et al., 2018). In 1877, a great famine occurred in southern India due to persistent drought (Buchan, 1877). By 1878, the drought spread further to Egypt, Morocco, the West Indies, Guyana, Venezuela, Colombia, and Brazil. The decline of canal water levels in some





countries has affected their foreign trade (Whitmee, 1878). Droughts have even affected the survival of regimes and civilizations. Indeed, the collapse of the Mayan civilization might have been caused by a drought that lasted for nearly 100 years in the 9th century (Gerald et al., 2003). In China, the climate drivers that led to the fall of the Dali Kingdom (937–1254) and the Ming Dynasty (1368–1644) were persistent droughts (Tan et al., 2018; Zheng et al., 2014).

In China, there is no lack of paleoclimate research using historical documents and materials. Based on official records and historical materials from the Qing Dynasty combined with local chronicles, it was found that the periods 1777–1778, 1876–1877, 1900–1902, and 1919–1920 were the years of the most extreme droughts in the last 300 years in North China (Zheng et al., 2018). Based on official records and local chronicles in the Qing Dynasty, it was possible to reconstruct the Dynasty's societal response to the agricultural failure caused by low temperatures and drought after the eruption of the Tambora

volcano in 1815 (Hao et al., 2020). The records of the annual final snowfall dates in ancient books allowed the reconstruction of the final spring snowfall in Hangzhou during the Southern Song Dynasty period 1170–1270. The variation in the final snowfall dates spanned 27 days, which was fewer than the 30-day range observed between 1951 and 2010 (Liu et al., 2017). According to the climate records in ancient diaries, the cold winter of 1308–1309 in the lower reaches of the Yangtze River in China might have been the trigger for the transition from the Medieval Warm Period (MWP) to the Little

Ice Age (LIA) (Chen et al., 2020).

Monsoons have unstable characteristics, and the speed and rhythm of the northward movement of the main rain belt of the East Asian summer monsoon fluctuate every year, which leads to interannual and inter-seasonal drought–flood fluctuations in China's agricultural areas (Wang and Lin, 2002; Hao et al., 2018; Hao et al., 2019). In 1876, the eastern part of the North China agricultural region suffered from a drought, and the summer rainfall was only half that of the 1971–2000 average. In

1877, a drought affected the eastern area of the Chinse Loess plateau and the entire North China Plain. The severe drought lasted until 1878, causing nearly 20 million deaths and migrations, with up to 3 million people migrating from Shandong Province alone (Hao et al., 2010; Ge et al., 2016). The severe drought during the period 1928–1930 caused more than 2 million deaths in Shaanxi Province (Zheng, 2001). As early as the 1620s, the Chinese Loess Plateau suffered from drought, resulting in the Gao Yingxiang and Li Zicheng uprisings, which eventually led to the fall of the Ming Dynasty (Zheng et al.,

2014). However, it is worth noting that the drought events that resulted in heavy loss of life all occurred during turbulent periods, when society could not effectively respond and adapt to disasters. While droughts that occurred during periods of favorable climate, relative prosperity and social stability were mostly resolved by strong societal responses, as was the case of the severe drought in Zhejiang in 1751 (Hao et al., 2021). By sorting out the grain harvest scores from the Qing Dynasty and combining this information with the natural disaster records in historical materials, it was found that, in 1759, a severe

drought event occurred in the agricultural areas of Northwest China. While many areas suffered a hundred-year agricultural failure, the 1759 drought event did not cause serious social unrest. It can be seen that the societal adaptability at that time resolved the negative impact of the drought-driven agricultural failure. This study reconstructs the spatial-temporal distribution of the 1759 drought, its severity, and the societal response measures using grain harvest scores and historical



records. We analyze and discuss how the agricultural society of the time adapted to a drought in the agricultural area of
Northwest China.

## 2 Study area and data

### 2.1 Study area

Northwest China covers an area of more than 3 million km2, accounting for nearly one-third of China's land area. It is in
China's inland, with diverse topography and landforms, most of which is characterized by arid and semi-arid climates (Liu,
2010; Zheng et al., 2013). At present, this region consists of five provinces, including Shaanxi, Gansu, Ningxia, Qinghai, and
Xinjiang. Due to the impact of summer monsoon, crops can be planted in the eastern part of Northwest China. This region,
along with four other provinces in the northern part of China, Hebei (also known as Zhili in the Qing Dynasty), Shandong,
Shanxi and Henan, formed the Northern Agricultural Region of ancient China (NAR) (Figure 1a). Shaanxi and Gansu
provinces, located in the eastern part of Northwest China (at that time, Gansu included present-day Gansu Province, Ningxia,
and the eastern part of Qinghai Province), were the Northwest Agricultural Region of China (NWAR) (Figure 1b). However,
Qinghai and Xinjiang provinces in the western part of Northwest China were mainly pastoral areas in ancient China due to
dry climate and little influence from summer monsoon. As an agricultural area, there were quantitative, score-based records
of agricultural yields for the NWAR in the Qing Dynasty, as evidence of crop abundance. Therefore this study focuses on the
NWAR. At that time, other areas in Northwest China were sparsely populated and were pastoral areas, which had little
impact on the overall economy and stability of the agricultural society.

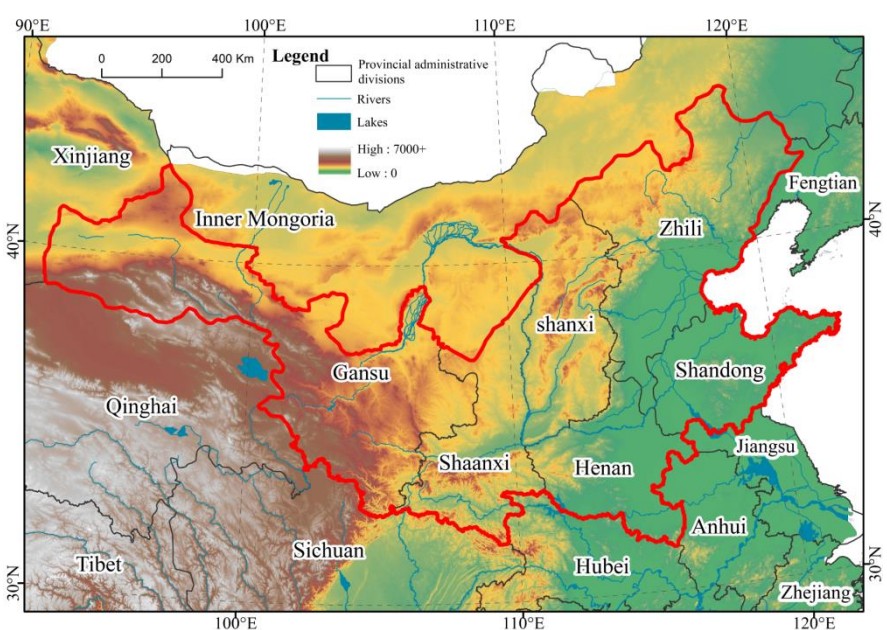



**a**

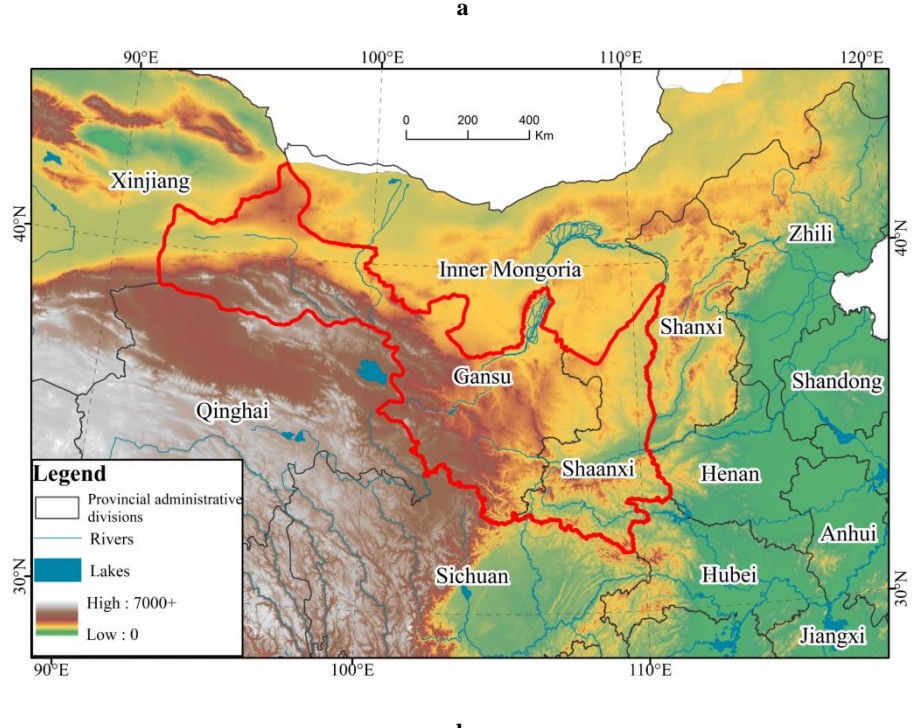

**b**

Figure 1. Outlines of NAR(a) and the two provinces of NWAR(b) during the Qing Dynasty. The provincial boundaries shown are those from the Qing Dynasty, drawn according to the Concise Historical Atlas of China (Tan, 1991).

### 2.2 Data sources

### 2.2.1 Historical data

The historical documents used in this research are two national-level official and authoritative documents about the history of the Qing Dynasty. The first is the Records of Qing Dynasty: Qianlong Biography (RQD) (Hanlin Academy of the Qing Dynasty, 1911), which records the national events of each year during the reign of every emperor of the Qing Dynasty, including a description of the location, duration, and severity of crop damage caused by major droughts and floods, as well as how the society rescued the affected areas. By using the second document, the Chinese Almanac for Two Thousand Years (Perpetual Calendar Editing Team, 1994), each Chinese lunar calendar date in the first historical document was transformed into the universal Gregorian calendar date. It is worth mentioning that the time involved in this study is mainly concentrated on the year 1759, and the main reference historical source is the RQD. Therefore, the records that do not specifically mark the year all occurred in 1759, and the records that are not marked with references are all from the RQD in order to be concise.



### 2.2.2 Agricultural harvest data

To know the harvest status of every agricultural province across the country in a timely and accurate manner and to adjust the taxation standards or relief efforts for areas with different levels of agricultural abundance, the Qing government established an agricultural harvest reporting system used during the period 1730–1910. According to this system, local provinces, prefectures and counties needed to report the local grain harvest within 1–2 months after harvesting. These reports are now preserved in the First Historical Archives of China, including the overall status of the province's summer harvest

(mostly winter wheat, rice, and barley) and autumn harvest (primarily rice, spring wheat, and corn), as well as the harvest scores of the administrative units (department, prefecture, subprefecture) under its jurisdiction. The harvest score is a quantitative description of the degree of agricultural harvest, commonly assessed using the 10-fen method (Hao et al., 2020). The general idea is that the higher the score, the better the harvest; the lower the score, the worse the harvest. For example, 10-fen indicates an unprecedented harvest; 7-fen indicates a normal harvest; and a score less than 7-fen indicates a poor

harvest related to crop failure and disaster (Zhang, 1996; Hao et al., 2021).

Among the 87 sites with harvest records in the Qing Dynasty, this study used 34 sites located in the NAR of the Qing Dynasty, including 8 sites in the NWAR which were in the western region of the NAR (yellow dots) and the remaining 26 sites adjacent to the NWAR but also within the NAR (green dots). It should be noted that although Hanzhong and Ankang sites belong to NWAR's Shaanxi Province, they were located south of the Qinling Mountains, which is an important climate

boundary. Due to the influence of different types of air masses, these 2 sites and the other 8 climate sites of NWAR had low consistency in climate, so the 2 sites were not considered typical NWAR sites for research and were classified as adjacent sites of NWAR (green dots). These 34 sites are representative of the overall grain harvest in the NAR (Figure 2). Furthermore, through the grain harvest score, the intensity of summer monsoon precipitation throughout NAR in 1759 can be inferred.




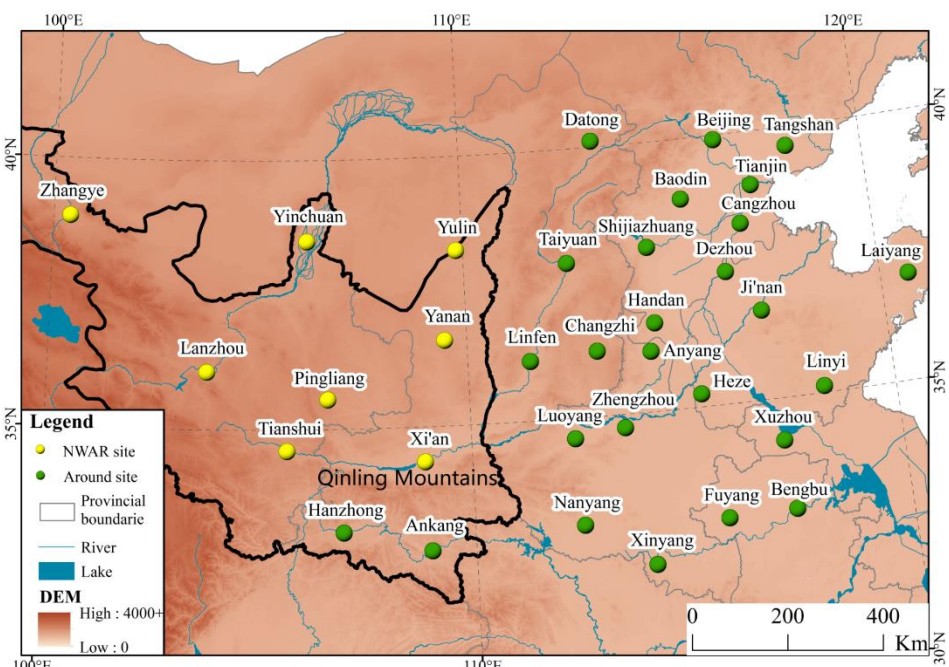

**Figure 2. Distribution of agricultural harvest sites in the NAR, including NWAR sites (yellow) and adjacent sites (green).**

Because a poor grain harvest is generally driven by climate disasters, the year of a poor grain harvest might be the year of an

extreme climate event. Therefore, the year in which a climate disaster occurred can be deduced from the annual fluctuation
in the grain harvest score at each site. By verifying with local chronicles and other relevant historical materials, we
determined the climate disaster type, reconstructed the process of this severe climate event and conducted research on
societal disaster relief measures.

Based on the agricultural harvest scores from eight NWAR sites in 1730–1910, a sequence diagram of the scores was plotted

(Figure 3). It can be seen that there was a significantly low harvest score of only 5.25 fen in 1759, which was the worst year
for agricultural harvests between 1730 and 1900. It was lower than the average harvest score (7.01 fen) of 1730–1910 by
1.75 fen. The figure also shows that the poor harvest in 1759 reached a once-in-a-century severity. By verifying with the
records in the RQD, there was no social unrest that interfered with agricultural production in the area where the eight sites
were located, but there were records of drought: "(26 May to 23 July) In Gansu Province, Gaolan and 36 other counties had

insufficient precipitation, and droughts appeared in various places. (26 May to 24 June) Shaanxi Province also suffered from
drought… The drought to the east of the Yellow River in Gansu Province led to a disaster in the summer harvest." The poor
harvest in this area in 1759 thus appears to have been caused by drought.





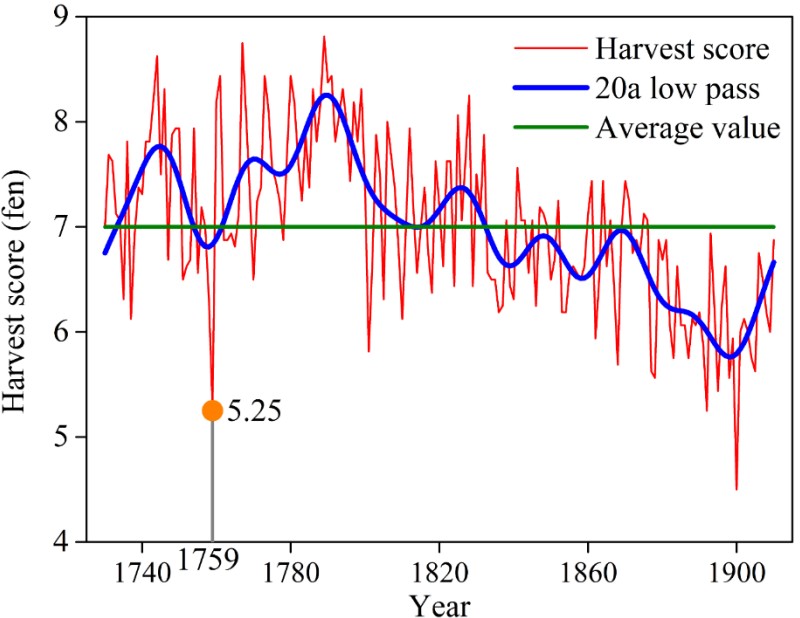

**Figure 3. Sequence of average agricultural harvest scores at 8 NWAR sites.**

### 2.3 Extraction and collation of historical materials

The texts on the drought records were extracted after a comprehensive reading of the records about the 1759 drought in the
RQD. Through the various types of records, it is possible to reconstruct the complete process related to the drought from the
occurrence of climatological drought to the impact of drought on crops, the impact of harvest failure on human society, the
response of human society to the impact of drought, and the full resolution of the drought impact in the end (Table 1).

**Table 1. RQD records related to the extreme drought event of 1759.**

| Record category | Meteorological records related to the northward movement time of the summer monsoon rain belt. |
|---|---|
| Evidence of late rainy season leading to drought | (26 May to 24 June) Less rain in Beijing and Shanxi … There has been no effective precipitation on both sides of the Yellow River in Gansu Province since spring, and the summer harvest may be poor … |
| Records of monsoon rain belt reaching the north | Since 5 July, Beijing had several heavy rains (24 July to 22 August). Since the beginning of summer, the Zhili Plain and the Taihang Mountains have had abundant rainfall, and the autumn crops may have a bumper harvest … (21 September to 20 October) The Zhenghe River, Weihai River, and other rivers had high water levels overflowing the embankments … Although Shaanxi Province had abundant precipitation this autumn, the farming season had passed, and some areas |





| | were affected by disasters. |
|---|---|
| Record category | Drought-affected areas and severity in agriculture. |
| Drought-affected areas | (26 May to 24 June) Thirty counties (including Xianyang, Chang'an and Wugong) around Xi'an, Shaanxi Province. Despite several rains since summer, they did not guarantee the growth of crops. / (26 May to 24 June) In Xunzou and areas east of the Yellow River in Gansu Province, the summer harvest suffered a failure. / (From 26 May to 23 July, there was no abundant precipitation in 36 counties (including Gaolan) in Gansu Province from 26 May to 23 July. Most of the crops were affected by drought, and food prices were high. |
| Severity within drought-affected areas | (20 November to 18 December) Despite the not so bumper crop harvest across 10 counties (such as Xianyang) around Xi'an in Shaanxi Province, it did not reach the disaster level/ (18 January 1760 to 16 February 1760) Yan'an and Yulin in northern Shaanxi suffered severe harvest failures and caused disasters. / (17 February 1760 to 16 March 1760) The agricultural harvest in Gansu Province was poor in the previous year. Among them, 10 counties (including Gaolan) were the most severely affected, 8 counties (including Jingning) were seriously affected, and 17 counties (including Weiyuan) were less affected. |
| Record category | Human societal aspect: Societal response to drought at the time |
| Records of tax exemption or postponement | (24 July to 22 August) Gansu has loaned rations (to be returned within the next 3 years) to the indigenous people along the border for three months after the poor summer harvest. / (23 August to 20 September) Stopped collecting taxes from people affected by drought this year in 39 counties (including Yangqu, Kelan, Baode, and Hequ) of Shanxi Province. / (21 September to 20 October) Took out the grain storage in Gansu Province to aid the residents of 40 counties (such as Gaolan, Jinxian, and Huamachi) who suffered from famine due to drought this year. / (17 February 1760 to 16 March 1760) Exemption of all agricultural taxes due in Gansu Province in 1760. |
| Records of food allocation to disaster areas | (26 May to 24 June) Rescued Shaanxi Province with rice from Sichuan Province. / (26 May to 24 June) Dispatched 0.2 million dans of grain from Shaanxi Province to Wuwei, Gaolan, and other counties in Gansu Province. / (25 June to 23 July) Allocated 0.2 million dans of grain from Jiangsu Province to Beijing. / (21 October to 19 November) Allocations of 0.05 million dans of grain from Shandong Province and 0.1 million dans of grain from Jiangsu Province arrived in Beijing. / (21 October to 19 November) Allocation of 0.04 million dans of stored grain from Chengxian County (Gansu Province) to Xihe County 95 kms away. / (17 February 1760 to 16 March 1760) Allocation of 0.1 million dans of grain storage from Yinchuan, Xining, and other places to disaster-stricken areas in Gansu Province. / (16 March 1760) The grain storage quota in Gansu Province was 3.7 million dans. After aiding the drought-affected people, the grain storage dropped to 1,375,622 dans. |



| | |
|---|---|
| Other disaster relief measures | (24 July) Seed grain was loaned to drought-stricken farmers in 36 counties (including Gaolan) in Gansu Province to replant the affected fields in time. / (23 August to 20 September) Six counties (e.g., Didao, Hezhou, and others) were affected by the disaster. The disaster-affected residents were employed to build the previously collapsed walls. They were paid labor wages to buy food. / (21 October to 19 November) The disaster-affected poor people in Gansu Province were hired to build water conservancy facilities (such as canals and dams) in Shandan County. / (20 November to 18 December) As Gansu Province lacked circulating currency to buy grain from other provinces, (the central government) ordered Sichuan and Hubei provinces to deliver 0.12 million and 0.08 million strings of copper coins, respectively, each year to help Gansu Province. / (17 February 1760 to 16 March 1760) The date for returning seeds and cattle previously lent to the famine refugees for spring ploughing was postponed until after the autumn harvest this year. |
| Disaster relief | (12 July 1760 to 10 August 1760) This year from spring to summer in Gansu Province, rain was abundant, crops were growing well, and food prices were falling. / (9 September 1760 to 8 October 1760) Shanxi Province had a good combination of precipitation and sunshine in summer and autumn, which resulted in favorable summer and autumn harvests. / (9 October 1760 to 7 November 1760) The areas hit by disaster last year, Yan'an, Yulin and Suide, all had bumper harvests this year. |

"dan" is a measurement unit for grain, 1 dan = 72.52–83.5 kg; 'tael' is a measurement unit for gold and silver, 1 tael of silver could buy about 50–60 kg of rice in Beijing in the 1740s.

## 3 Results

### 3.1 Rainy season in the NAR in 1759

From 26 May to 24 June 1759, almost the entire NAR was dry, with little rain. From east to west, "rain did not arrive on time" near Beijing in the central part of Zhili Province, and there were also records of lack of rain in many counties in southern Zhili Province. Due to the lack of precipitation in Shanxi Province from 26 May to 24 June, officials even went to the temples on Mount Wutai to pray for rain. Despite several light rains in more than 30 counties near Xi'an in Shaanxi Province, precipitation failed to meet the water demand for crops. In Yan'an and Yulin in northern Shaanxi, the winter wheat harvest was a failure due to drought. Similarly, there was little rain in the westernmost part (both sides of the Yellow River in Gansu Province and in the vast area of the Hexi Corridor) of the NAR. Thus, it can be seen that from 26 May to 24 June, drought emerged across the entire NAR.

From 25 June to 23 July, with the northward movement of the monsoon rain belt, the Guanzhong Plain in Shaanxi Province received some effective rainfall, which alleviated the drought, and the harvest of winter wheat was good. After July 5, the



Haihe Plain in Zhili Province also entered the rainy season. After several heavy rains, the autumn crop harvest was secured in Zhili Province. Nonetheless, at that time, the main monsoon rain belt did not spread further to Shanxi and the northern

Shaanxi provinces, and the whole Gansu Province had no effective precipitation for crops until 23 July. It can be seen that in June, the rain belt moved northward to the Guanzhong Plain in Shaanxi Province, and the low-altitude southern parts of Zhili Province, while the drought conditions continued to worsen in Shanxi, northern Shaanxi and Gansu provinces, which were located in mountainous regions and the Loess Plateau.

On 24 July, Shanxi Province entered the rainy season. For instance, heavy rains fell from 24 July to 22 August near the

provincial capital Taiyuan, and excessive water flow from the upper reaches of the Haihe River in the Taihang Mountains of eastern Shanxi (in the upper reaches of the Haihe River) caused minor floods in the Haihe River Plain in Zhili Province (in the lower reaches of Haihe River). After 24 July, Gansu Province entered the rainy season and had more abundant precipitation. It can be seen that from 24 July to 22 August, with the northward movement of the monsoon rain belt, the entire NAR appeared to enter the rainy season, marking the end of the drought in meteorological terms.

By 23 August, under the influence of the main rain belt of the summer monsoon, the accumulated precipitation in some areas of the eastern plains in the NAR became excessive. From the spring–summer drought to the summer–autumn flood, the most obvious change was in Zhili Province in the east. Zhanghe River, Weihe River, Daqinghe River, and Baiyangdian Lake had high water levels. Some villages around the rivers and lakes were inundated due to the collapse of dams. By 21 September, all departments and prefectures in the southern part of Zhili Province were lightly flooded.

**3.2 Spatial and temporal distribution and severity of the drought**

The lack of rain in the spring and summer of 1759 led to droughts of different degrees in the NAR. Zhili Province suffered from a drought in June, and eight counties (e.g., Qingyuan County) experienced poor summer harvests. Due to insufficient precipitation before 24 June, Shanxi Province was expected to lose the summer harvest and requested grain relief from Henan Province. From 25 June to 23 July, the province indeed suffered from a severe drought. Yan'an and Yulin in the

Loess Plateau of northern Shaanxi Province had a poor winter wheat harvest in summer due to the lack of effective precipitation. It made Shaanxi Province difficult to rescue the drought-stricken Gansu in the west, which had insufficient precipitation from 26 May to 23 July, indicating an upcoming severe crop failure. It can be seen that before 25 June in 1759, the NAR suffered from extensive drought due to inadequate rain in spring and summer, resulting in a general failure of the summer crops. After 25 June, as the summer monsoon rain belt reached the Haihe Plain in the central and southern parts of

Zhili Province and the Guanzhong Plain in Shaanxi Province, the drought conditions in these two places were relieved. Specifically, heavy rains after 5 July sufficiently satisfied the water requirements for autumn crops in Zhili province. From 25 June to 23 July, the Guanzhong Plain and its surrounding areas in Shaanxi Province had abundant rainfall. Xi'an, Baoji, and other sites were expected to have a bumper autumn harvest.

However, the drought conditions in other areas of the NAR continued, and increased in severity. For instance, "(Shanxi

Province) Until 24 July, there was still no sufficient precipitation near Taiyuan, and wheat harvests were poor in Yulin and





other places in northern Shaanxi, while 36 counties in Gansu Province had no effective precipitation until 24 July. The summer crops were damaged, and food prices were high." After 24 July, the drought conditions in the above areas gradually eased. In farmland from the Datong area of Xinzhou in northern Shanxi to the north of the Great Wall, autumn grain crops recovered slightly due to increased precipitation. Nonetheless, with the previous long-term drought, the poor harvest was a foregone conclusion. In Gansu Province, all areas received abundant precipitation from 24 July to 20 September, which irrigated crops that had suffered from severe drought earlier. The arrival of rainwater in north of Shaanxi Province in autumn was too late. The autumn crops in these areas thus had serious failures. It can be seen that, despite the alleviation of the drought in Shanxi and northern Shaanxi and Gansu provinces in autumn, the severe drought from late spring to the midsummer caused irreversible damage to crop growth and caused the low grain harvest in the NWAR this year.

The plain area of Zhili Province and the Guanzhong Plain of Shaanxi Province entered the rainy season from 25 June to 5 July. Among the spring–summer drought-affected areas in the NAR, they were the first to enter the rainy season. Thus, the degree of drought between the two should be almost the same, and they should be among the areas only mildly affected by drought. Because the rest of Shanxi and northern Shaanxi and Gansu provinces only received effective precipitation after 24 July (i.e., 30 days later than the former two places), they experienced relatively severe impact from the drought. There are statements in the RQD that can directly or indirectly reflect varying drought severity in different areas (Table 1). In this study the differing severity of the disasters in the affected areas is clearly distinguished. It is also possible to distinguish the degree of drought impact by referring to the degree of relief efforts in the drought-affected areas. For example, as presented in Table 1, the disaster situation in the area receiving relief for two months is obviously more serious than that in the area receiving relief for only one month. Based on these records, we can clarify the differing severity of the disasters within the NAR drought-affected areas (Figure 4).





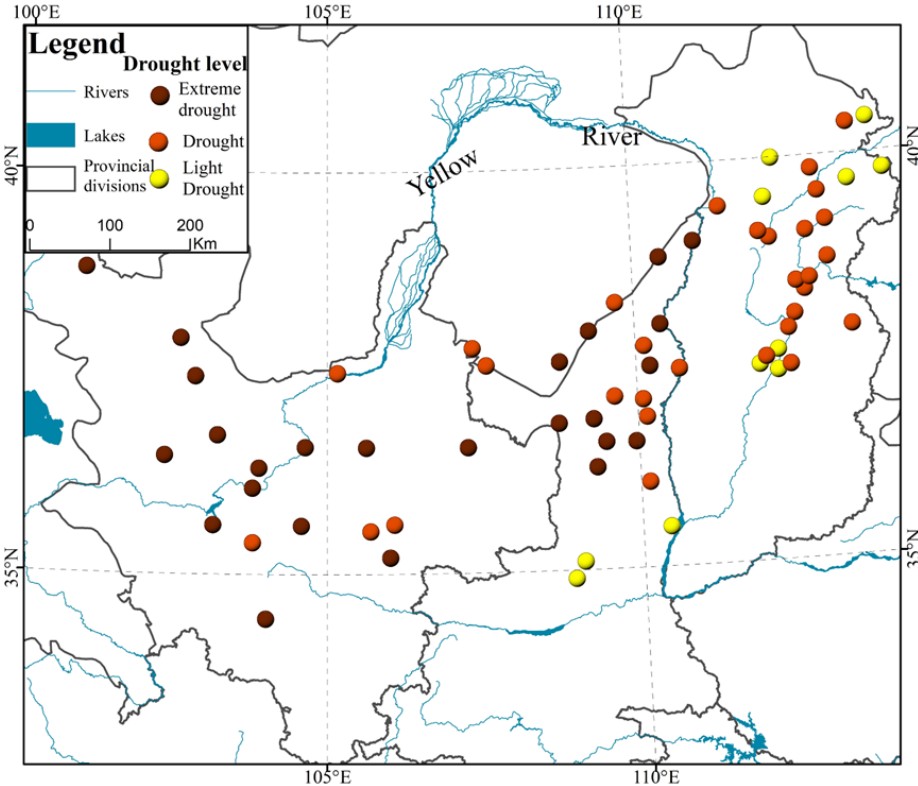

**Figure 4. Distribution map of drought severity at recorded sites in the NAR in 1759 (drought conditions in the Guanzhong Plain and Zhili Province were less severe than all the sites marked in colors in the figure, so they are not shown in the figure).**


To further explore the drought severity in the different areas of the NAR in 1759, this study applies the data of grain harvest scores. The lower the crop harvest score, the more severe the drought (as shown in Figure 5a, b). Due to the lack of rain in the NAR during spring and summer, the summer harvests, which relied mainly on spring rain, all had very low scores. Among them, the worst harvest sites were Baoding and Cangzhou in Zhili Province and Lanzhou, Yinchuan, Tianshui, and

Pingliang in Gansu Province, all of which got 4 fen. Seven sites (including Tianjin, Datong, Yan'an, and other sites) had 5 fen. The sites with the worst autumn harvest scores were Lanzhou, Pingliang, Datong, Taiyuan and Linfen at 5 fen; Cangzhou, Tianjin and Beijing in Zhili Province achieved 7 fen; while Baoding had bumper harvest at 8 fen. Overall, in the NAR, the autumn harvest scores were higher than the summer harvest scores, which corresponds to the meteorological records that there was a lack of rain in spring and summer in northern China in 1759 and that the summer monsoon brought

rain belt northward and eased most of the drought conditions in NAR after midsummer.

For the annual average harvests (Figure 5c), Gansu, northern Shaanxi and Shanxi provinces were the most severely affected, corresponding to the severely affected areas recorded in the RQD. Lanzhou and Pingliang in Gansu Province had only 4.5 fen; while Yinchuan and Tianshui in Gansu Province and Taiyuan and Datong in Shanxi Province scored 5 fen. These were the sites with the worst annual average harvest scores in 1759. According to the average grain harvest sequence of all sites





north of Qinling Mountains in Gansu, Shanxi, and Shaanxi provinces during the period 1730–1910, these regions experienced a harvest failure of once-in-a-century severity in 1759 (at 5.25 fen), which was even more serious than the worst year of 1877 (at 5.56 fen) of the Ding-wu Disaster in 1876–1877 (Figure 3). In the entire NAR, south of the Qinling Mountains in Shaanxi, Henan, Shandong, and the northern part of Anhui and Jiangsu, adjacent to the drought-affected areas, had better agricultural harvests in 1759 (Figures 5a-c). This confirms the RQD's records of drought-stricken areas and

bumper-harvest areas. The bumper harvests areas strongly supported the central government's relief efforts in the drought-stricken areas that year.

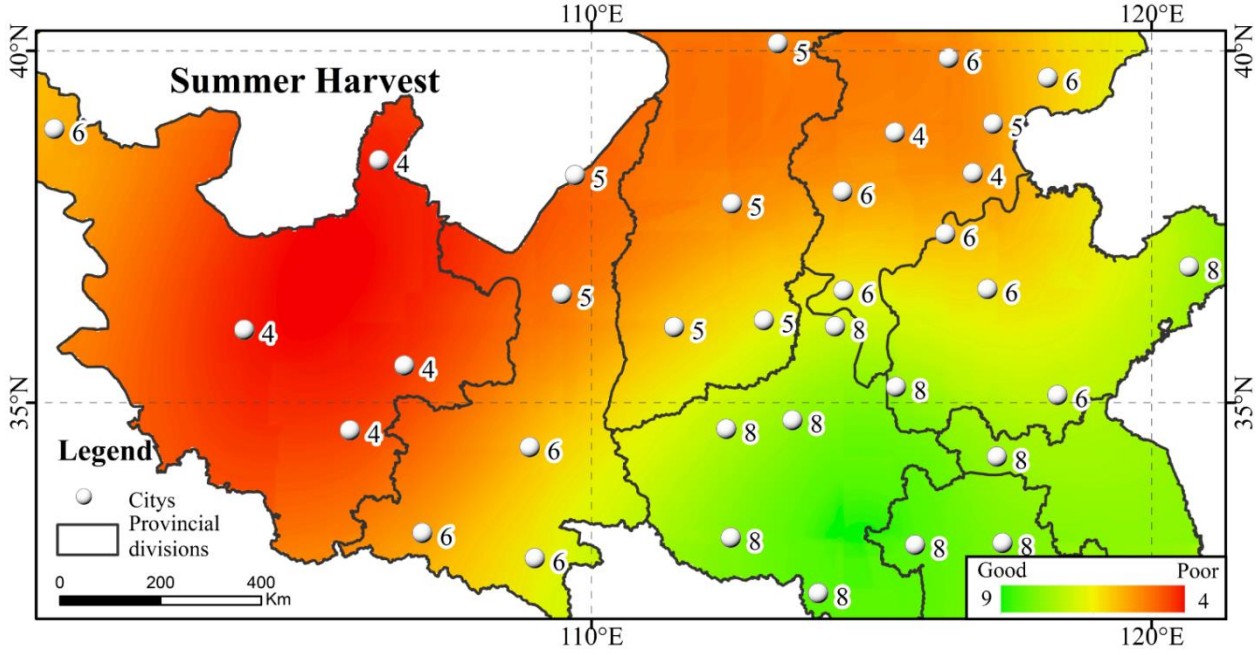

a



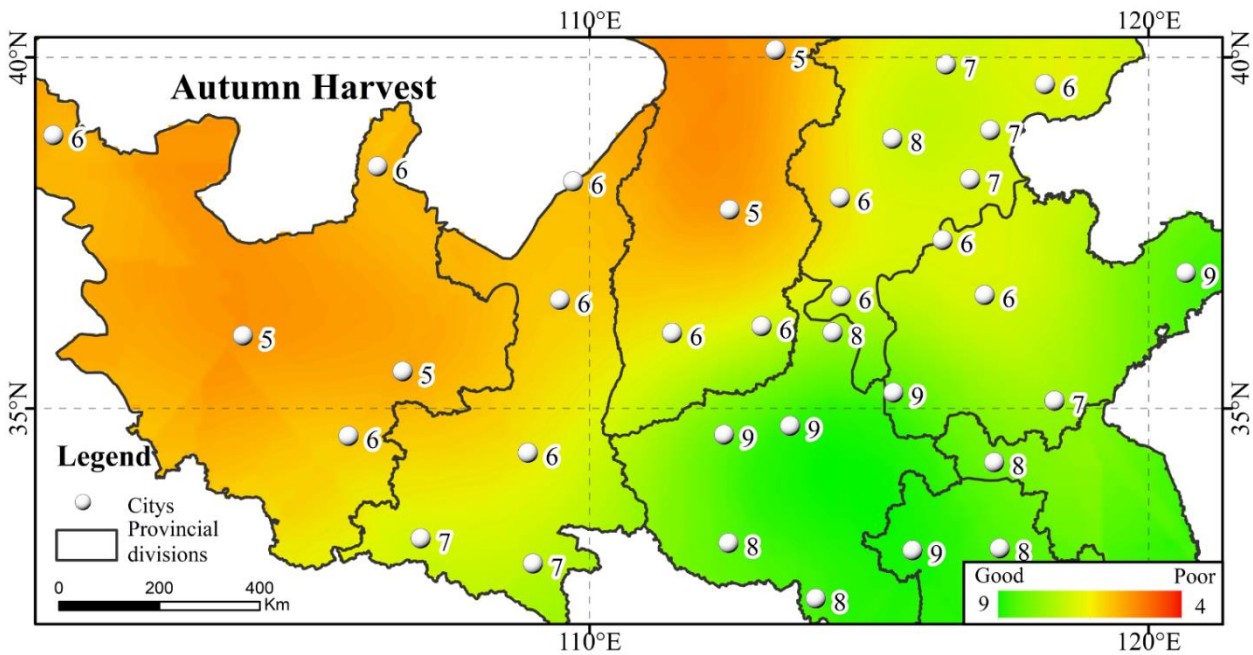

245                                                                b

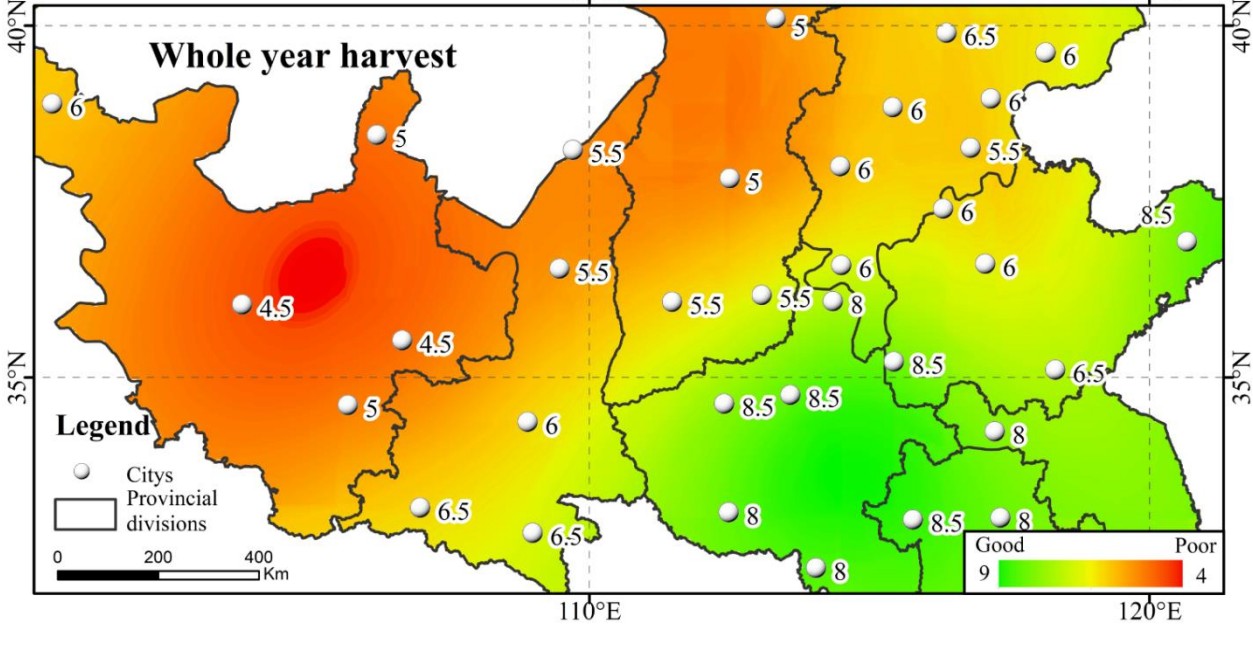

c

**Figure 5. Summer harvest (a), autumn harvest (b), and average summer and autumn harvest (c) at NAR sites in 1759.**



### 3.3 Societal response to drought

In ancient Chinese agricultural society, there were three major relief measures for disaster-stricken areas: tax exemption for the disaster-stricken areas, using warehouses to store grain for disaster relief, and grain allocation from harvest areas to disaster-stricken areas.

Exemption or postponement of tax collection was the most common disaster relief measure. When the 1759 drought occurred, the central government immediately announced on 24 July that the tax collection for the year in 25 prefectures and counties would be postponed. Similar policies were also issued for the severely affected Gansu Province and northern Shaanxi Province. For example, from 21 October to 18 December, the central government deferred tax collection from the drought-affected people in Gansu Province, and areas surrounding Yan'an and Yulin prefectures in Shaanxi Province, from 1759 to 1760. Despite the implementation of similar measures in almost all drought-affected areas, the severe drought rendered tax exemption alone inadequate to bring food directly to the people. Other more effective disaster relief measures were also required.

To cope with the rising grain prices caused by fluctuations in crop yields, in 1748 the Qing Dynasty established a quota for grain storage in government warehouses for each province to prepare for potential famine. These records are available in another authoritative historical document of the Qing Dynasty, Annals of Food and Goods from the History of the Qing Dynasty (Zhao, 1927). The designated quotas for Zhili, Shanxi, Shaanxi, and Gansu provinces were 2.1 million dans, 1.3 million dans, 2.7 million dans, and 3.7 million dans, respectively. The grain stored would be taken out to support refugees in the event of serious disasters. In Gansu Province, from 17 February to 16 March 1760, the government took out more than 0.2 million dans of grain from warehouses in Xining, Yinchuan, and other places for disaster relief around them. Different areas in the same province could also transfer grain to each other. For example, Chengxian County in southern Gansu, where the disaster was less severe, took out 0.04 million dans of grain from the warehouse on 19 November to help the more severely affected counties around Lanzhou in the province. This stored grain, in addition to providing rations for the victims, could also provide seeds for the next year to facilitate the following year's production in the disaster area. For instance, the government provided the seeds for spring ploughing in 1760 to disaster victims in Gansu Province. After the disaster relief in 1759, the grain in the Gansu provincial warehouse declined by nearly 2.3 million dans from the quota of 3.7 million dans, leaving only 1.376 million dans of grain (i.e., 37.19% of the quota). However, the drought in 1759 was the worst drought in a century, and stored grain in the affected provinces was not sufficient for disaster relief. The central government thus initiated and coordinated inter-provincial grain allocation.

As early as May, when the drought first occurred, the Qing government ordered Shaanxi Province to transfer 0.2 million dans of grain to warehouses across Gansu Province to prevent the disaster from worsening. On 17 February 1760, Shaanxi Province again transferred 0.4 million dans of grain to Gansu Province to help areas hit by the 1759 drought. In addition to Shaanxi Province, Sichuan Province in southern China also rescued the disaster-hit areas in Gansu Province. For example, from 26 May to 24 June, the central government dispatched Sichuan grains to Lueyang in Shaanxi, which were then





transferred to Gansu (Fig. 6). These disaster relief grains from other provinces could be put into the grain market in the disaster-stricken areas to stabilize grain prices, as well as providing direct and effective relief to famine refugees by opening

food relief centers in the hard-hit areas. For example, in early 1760, the disaster relief grains from Shaanxi to Gansu was sent to a food relief center, which directly helped the starving refugees.

Another disaster-stricken province, Shanxi, received food aid from Henan Province from 26 May to 24 June, which was transferred via the Yellow River to the Fen River. Henan Province was able to help Shanxi because of its bumper agricultural harvest in 1759 (Figure 5c). Zhili Province, the easternmost drought-stricken area, was aided by both Shandong

and Jiangsu provinces; Beijing, for example, shipped 0.05 million dans of grain from the bumper harvest of southern Shandong from 25 June to 23 July. On 19 November, Shandong and Jiangsu shipped 0.05 million and 0.1 million dans of grain, respectively, to enrich the Beijing grain market. The success of the above-mentioned grain transfers from different regions benefited from the bumper harvests in Shandong, Henan, northern Jiangsu and Anhui in 1759 (Figure 5c). These provinces were the main grain production areas in China, and they strongly supported Zhili and Shanxi in 1759. The disaster

relief in the two provinces reduced the pressure on the central government to provide relief to Gansu Province (the hardest-hit province), and indirectly supported the disaster relief in Gansu Province. Notably, although the northern part of Shaanxi Province was also affected by drought, the monsoon rains arrived in time in Guanzhong Plain around Xi'an in central Shaanxi, and the autumn harvest score was slightly higher (6 fen). Moreover, Ankang and Hanzhong, south of the Qinling Mountains in Shaanxi Province had bumper autumn harvests (7 fen). Thus in the autumn of 1759, the warehouses around

Xi'an still contained 0.84 million dans of grain, which not only stabilized the grain prices in Shaanxi Province, but also helped the severely affected Gansu Province. The grain allocation routes to deal with the 1759 drought became clear, and the inter-provincial grain allocation routes were: Shaanxi and Sichuan aided Gansu; Henan aided Shanxi; and Jiangsu and Shandong aided Zhili. The three disaster-stricken provinces (Gansu, Shanxi, and Zhili) received aid from the longitudinally aligned southern provinces (Figure 6).



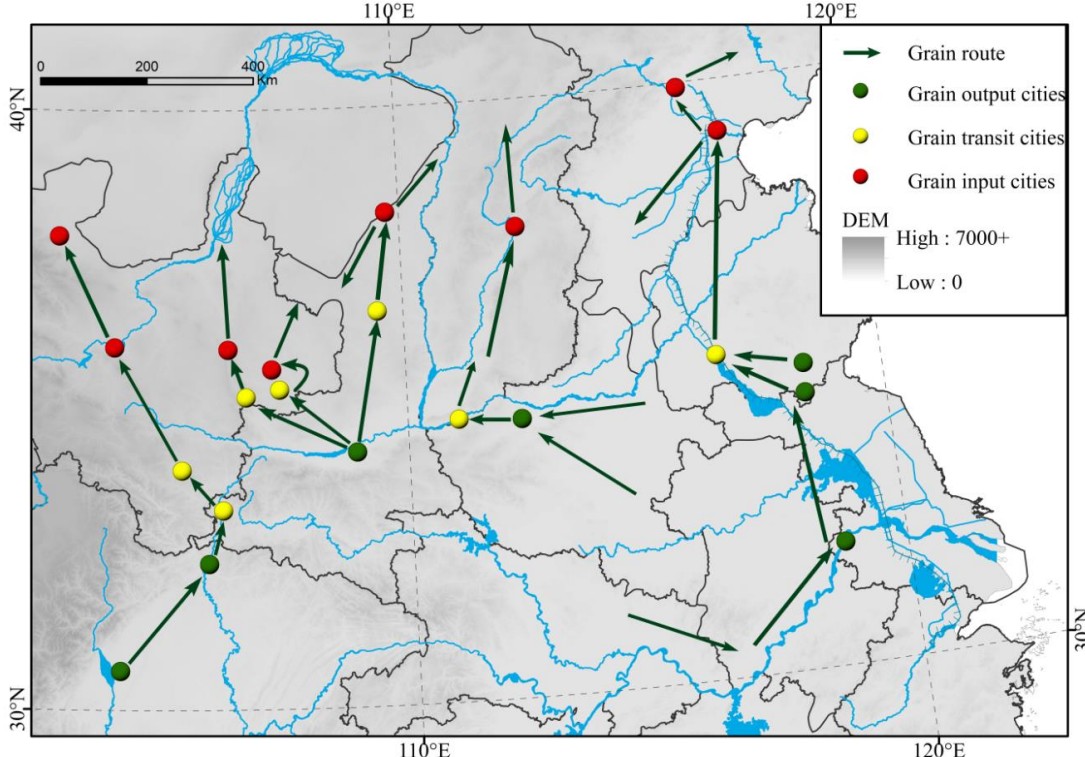

**Figure 6. Grain aid map for drought in northern China in 1759**

### 3.4 Disaster mitigation

By implementing measures such as tax exemption for disaster-stricken areas, using of stored grain in disaster-stricken areas to help refugees, and allocation of disaster-relief grain across provinces, the drought impact on the society was successfully resolved. According to the record in Chinese Three Thousand Years Meteorological Record Collection (Zhang, 2013), for Yongdeng County, Gansu Province, which suffered from severe drought, (1759) despite the poor agricultural harvest in the year, people did not go without food for long; for Lintao County, Gansu Province, during the severe drought in 1759, the superiors ordered the distribution of stored grain to help the refugees, and in the end, almost no one died from starvation. The drought impact ended in the summer of 1760 because of the bumper summer harvests in Gansu, Shanxi, and northern Shaanxi provinces. This year (1760) witnessed abundant rainfall, as a result, crops grew well, and grain prices became affordable, so relief measures were stopped on 13 June 1760. In Shanxi Province, counties such as Pingyang, which were hit by drought in 1759, had a bumper summer harvest in 1760. The whole province had abundant rainfall and sunshine in 1760, and enojyed bumper summer and autumn harvests. Yan'an, Yulin, and Suide in northern Shaanxi also had bumper harvests in 1760. By June 1760, drought situation in all of the hardest-hit areas in the NAR eased. From the beginning of the drought on 26 May 1759 till June 1760, the whole process lasted for one year, which was also the complete period from the occurrence of drought in meteorological terms to the time when the drought impact on the society was completely resolved.



Among other drought-affected areas, the Guanzhong Plain of Shaanxi Province had a bumper autumn harvest in 1759. Society was stable because of the timely arrival of rain. For eastern Zhili Province, although the price of grain rose slightly in the autumn of 1759, with the allocation of grain from other provinces, the people did not go hungry. The timely movement of the summer monsoon rain belt to the north resulted in a bumper autumn harvest in Zhili Province, and the drought conditions eased in September. Food prices around Beijing also fell back to normal. It can be seen that the impact of the drought on the society in the Guanzhong Plain around Xi'an in Shaanxi Province and the whole Zhili Province were already alleviated in the autumn of 1759.

## 4 Discussion

### 4.1 Other measures to assist disaster relief

In addition to measures such as tax exemption and transfer of grain, other disaster relief measures also played a role in the NWAR's successful response to the drought of 1759. These additional measures included: lending cattle to the disaster victims, balancing grain in the north and south through water transportation, work relief, increasing the wages of workers who transport grain to the disaster areas, and importing copper coins to the disaster-affected area to support them to buy grain from unaffected area.

In the early summer of 1759, to help farmers with their work, the government bought cattle and lent them to farmers in Yongchang, Gulang, and other counties in Gansu Province. These cattle helped replant the disaster-affected farmlands after the rains came. Furthermore, the drought relief in Zhili Province in northern China benefited from the use of "Cao rice" (i.e., grain transported annually from southern China to Beijing to pay officials and soldiers) for disaster relief. By 25 June 1759, Zhili Province had received a total of 0.8 million dans of Cao rice transported from the southern provinces. The timely support of Cao rice enriched the grain storage in Zhili Province, stabilized grain prices, and relieved pressure on the central government's disaster relief measures in Gansu Province (the hardest-hit area in the 1759 drought).

When drought occurred in Gansu Province, to relieve the disaster, the government hired the affected people in its nine counties (including Didao, Hezhou, Jingning, and Huanxian) to repair the dilapidated city walls and paid them to ensure that they could buy food. This was a common disaster relief measure in ancient China during periods of climate disaster. At the same time, to encourage the delivery of disaster relief provisions to Gansu Province, the government increased the labor fee for grain porters. The labor fee for grain transportation to Gansu Province was increased from 0.16 tael of silver · dan/50 km to 0.2 tael of silver · dan/50 km, with an additional payment of 0.06 tael of silver · dan/50 km for the return journey. Moreover, to facilitate the affected provinces purchasing grain from the provinces with bumper harvest, on 20 November 1759, the Qing government requested that Sichuan and Hubei provinces contribute to Gansu Province with 0.12 million and 0.08 million strings of copper coins, respectively, totaling 0.2 million strings of copper coins (about 0.2 million taels of silver in total) to enable Gansu Province to buy grain. After the disaster, the central government asked Sichuan and Hubei





provinces to fund Gansu Province with 0.2 million strings of copper coins each year to support its development and construction.

In ancient China, relief measures for droughts in different geographical regions were slightly different. For example, in Yunnan Province, a mountainous area in southwest China, the relief policy for meteorological drought-stricken areas also included increasing the construction of bridges between canyons to reduce the cost of grain transportation (Hao et al., 2020). In Zhejiang Province (with a relatively developed commodity economy in the southeast coast), grain traders were exempt from commodity tax to increase the share of grain imported into disaster areas (Hao et al., 2021). While in the NWAR, the

government lent cattle to farmers in disaster-stricken areas to facilitate timely replanting after the 1759 disaster, transferred copper coins from other provinces to help the disaster-stricken areas to buy grain, and increased the transportation service fee for grain porters to cope with the drought, which were different strategies than those used in other areas of China.

## 4.2 The impact of drought on society at the time

During the period 1755–1759, the Qing government carried out military operations against the vast area in the north and

south of the Tianshan Mountains to the west of NWAR (now Xinjiang Province). Since the NWAR was the main rear base to support the war in Xinjiang, the Qing government stopped any further military operations when the severe drought hit the northwest in 1759, and accelerated immigration to Gansu Province. This accelerated the development of Gansu and increased the Qing government's attention to the construction of water conservancy in Gansu.

By 1759, the territory of the Qing Dynasty expanded further to Kashgar, Hotan and Badakhshan in the eastern part of the

Pamirs (Badakhshan: now the Badakhshan Province in eastern Tajikistan and Wakhan Corridor in northeastern Afghanistan). The Qing Dynasty stopped the military operation in 1759 most likely because of the drought in the same year. At the beginning of 1759, because of the support for military operations, grain prices in the NWAR (as a logistics base) were already high. Most of the NWAR belongs to arid and semi-arid areas, with limited crop yields (Figure 7). This greatly restricted the military supplies for the Qing Dynasty's military operations in Xinjiang. The support for large-scale military

operations had been inadequate even during harvest year, and the 1759 drought finally made it impossible for the Qing Dynasty to support the military operations in Xinjiang logistically. After the drought, the Qing government stopped military operations in Xinjiang and immediately adopted a policy of recuperation in Gansu Province, such as exempting the entire province from the 1760 tax.



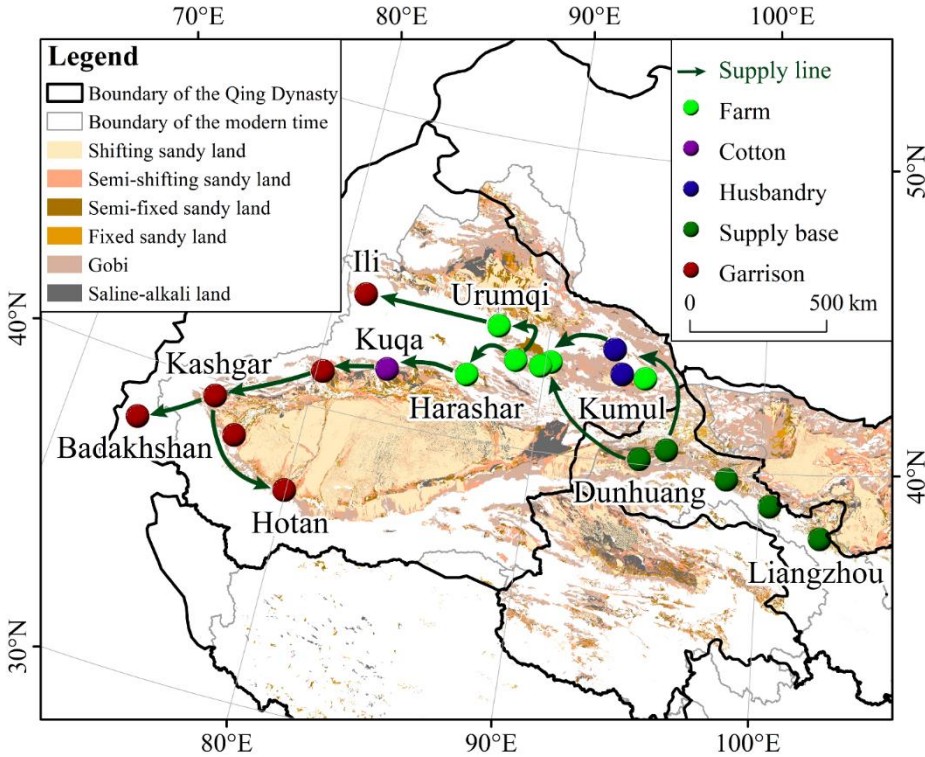

**Figure 7. Distribution of functional sites in Xinjiang in the middle Qing Dynasty.**


The drought in 1759 also had a negative impact on the stability of Xinjiang which was newly occupied by the Qing Dynasty. For this reason, the Qing government undertook a series of measures to help agricultural and animal husbandry production in Xinjiang. As early as 1758, the Qing government started to open up wasteland and grow grain in eastern Xinjiang, such as

Harashar, and harvested 0.037 million dans of grain that year, which met the local needs and also created a surplus for storage. In 1759, the Qing Dynasty opened up more than 5,000 mu (1 mu=0.0667 hectares) of arable land in Harashar and Urumqi, and tried planting peas, which yielded bumper harvests. With the expansion of reclaimed land area in Urumqi and other areas in northern Xinjiang year by year, the number of farmers migrating from other provinces gradually increased. This supported the reclamation of arable land and increased the grain output in Xinjiang. In terms of economic crops, the

Qing Dynasty shipped 200 catties of cotton seeds from Shaanxi to Kuqa and other places in Xingjiang in 1759 to develop cotton planting. Regarding animal husbandry, in 1760, in the pastures of Kumul and other places, the number of lambs bred reached more than 40,000; these sheep were sold to Gansu and Qinghai provinces while also meeting local demand (Figure 7). After the drought in 1759, the above measures not only maintained food supplies to the troops stationed in Xinjiang, but also allowed agriculture and animal husbandry in the newly war-torn Xinjiang to revive, laying the foundation for

development of Xinjiang after 1759.





Throughout the mid-18th century, the NWAR underwent great development, especially Gansu Province. In 1724, at the early development stage, the registered population of Gansu Province was only 302,800. By 1749, due to the development of agriculture, the registered population rose to 4,399,100. At that time, the province had stable grain production and abundant grain storage, and became the base for the Qing government to conduct military operations in the western Xinjiang. During the period 1771–1776, the population grew further from 10.5821 million to 12.0865 million (Wei, 1990). The rapid growth in population benefited from the society's greater resilience to climatic disasters (e.g., drought) in the 18th century. After the drought in 1759, the Qing government also paid more attention to the construction of water conservancy facilities in Gansu Province; for example, a dedicated official position responsible for water conservation construction was established in Gansu Province to strengthen the construction of irrigation canals and water conservation facilities in the region, which further promoted agricultural development in the province.

### 4.3 Long-term climate background and societal adaptation to climate disasters

In terms of the climate background, the successful response to the Gansu drought in 1759 benefited from the fact that, in the 18th Century, China as a whole was at a relatively warm stage of the Little Ice Age. The higher temperatures, compared with those in the 17th and 19th centuries provided climatic support for crop growth (Hao et al., 2011; Hao et al., 2012). While the droughts happened in the NAR in the 17th and 19th century caused strong social unrest. For instance, the 1638–1641 drought was the direct cause of the collapse of the Ming Dynasty (Zheng et al., 2014), and the 1876–1878 drought caused millions of deaths and tens of millions of migrations, thus severely affecting the socioeconomic situation at that time (Cook et al., 2010; Hao et al., 2010; Hao et al., 2021). Meanwhile, 1759 was only a year of unexpected harvest failure due to drought in the context of a century-scale harvest, while 1638–1641 and 1876–1878 were years of successive harvest failures due to prolonged drought in the contest of a century-scale harvest failure which significantly weakened society's ability to adapt to climate disasters.

### 5 Conclusion

This research draws three main conclusions. Firstly, from spring to summer in 1759, the NAR experienced a severe drought due to the continuous low rainfall in spring and summer, where Shanxi and northern Shaanxi and Gansu provinces did not receive effective precipitation until 23 July, leading to severe drought. By contrast, Zhili Province and the Guanzhong Plain of Shaanxi Province had abundant precipitation because of the monsoon rain belt moving northward from late June to early July, so the disaster was relatively mild in those areas. Affected by drought, the NWAR suffered a severe harvest failure in 1759, which was the worst harvest year in 1730–1900.

Secondly, in response to the severe drought in the NWAR, the Qing government adopted a series of disaster relief measures, such as tax exemption, the use of stored grain in the disaster-stricken areas, and the transfer of grain from other regions into the disaster area. These were accompanied by additional measures, such as lending cattle to the affected refugees to enable





timely crop planting, increasing the wages of workers transporting grain to the disaster area, and importing copper coins to the disaster area to support the purchasing of grain from other provinces. With the bumper summer and autumn harvests in the NAR in 1760, the influence of the drought ended; the impact of the 1759 drought on the society subsided smoothly and

did not escalate into more serious social unrest. When the drought occurred, the government stopped military operations in Xinjiang due to its limited ability to provide logistical support. After the drought, society paid more attention to the construction of water conservancy facilities in Gansu Province.

Thirdly, the climatic background in which the impact of the 1759 drought was quickly subdued was that, in the 18th century, China as a whole was in the relatively warm stage of the Little Ice Age in the Ming and Qing Dynasties (1368-1911 AD),

which enabled the Qing government to have sufficient financial resources and grain storage. Compared with the colder 17th and 19th centuries, the drought in the 18th century NAR did not lead to large-scale social unrest. This illustrates the impact of long-term climate fluctuations on the ability of a society to respond to climate disasters.

## Declarations

**Data availability.** Crop harvest data is available from the corresponding author upon reasonable request.


**Author contributions.** Zhixin Hao developed the research method, supervised the research of this study and collected crop harvest data. Haonan Yang defined the outline of this manuscript, Meirun Jiang drafted the manuscript, Danyang Xiong performed the calculations and most of the analysis, and drew the figures. Jingyun Zheng revised crop harvest data. All authors participated in the analysis, provided critical feedback and helped to shape the final manuscript.


**Competing interests.** The authors declare that they have no conflict of interest.

**Acknowledgements.** This research was supported by grants (to IGSNRR) from the National Natural Science Foundation of China (41831174).

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
