# Peer review of "Reconstruction of ancient drought in Northwest China and societal responses: A case study of 1759"

_Climate of the Past, 2023_

## Author Comment (AC1)

Dear Professor Fang,

We extend our sincere gratitude to you for your invaluable comments and constructive suggestions. Having thoroughly reviewed your feedback, we have gained valuable insights and identified areas where improvements were necessary. Notably, we have taken the initiative to address several deficiencies, including the revision of Figure 1a and b, which have been amalgamated into a single figure. Additionally, we have meticulously revised the aspects related to the concepts of time and space, as highlighted in your comments. We have also rectified other inappropriate elements that you brought to our attention.

It is indeed a privilege for us to have had the opportunity to receive your comments and suggestions. Below, we provide a comprehensive point-to-point account of the specific modifications made, with corresponding updates within the original paper. Thank you for your invaluable guidance.

**Professor Fang, 01 Aug 2023**

*This study presents a comprehensive reconstruction of the 1759 drought in NWAR, from the spatial-temporal distribution, to the impacts of the drought, and the societal responses. The historical records in NAR are arranged well to illustrate the drought process. The research draws three main conclusions, that is from spring to summer in 1759, the NAR experienced a severe drought due to the continuous low rainfall in spring and summer, the Qing government adopted a series of disaster relief measures to response the severe drought in the NWAR and the long-term climate fluctuations impact on the capacity of a society to respond to climate disasters. The results are valuable for disaster prevention, reduction and relief in modern society.*

*The title of "4.2 The impact of drought on society at the time", could not cover the contents of the part. The impacts of the drought were beyond the time and out of the region. It is suggested to change the title and focus the topic on the subsequent long-term impacts and the impacts out the connected regions.*

A: Suggestion taken. We revised the title according to the actual time and space mentioned in Section 4.2 to: "The impact of the 1759 drought on a larger spatial and temporal scale".

*1. Line 27. "Asia, Brazil, and Africa" are different scale units. Did only Brazil in South America suffer from the drought in 1877-1878?*

A: Thank you for your reminding. We reviewed and studied the article cited here again, and modified the area involved in Line 27. The original sentence has been changed to, "Historically, simultaneous multi-year droughts in China and India in Asia, Southeast Africa, and Northeast

Brazil in South America during the period 1875–1878 caused widespread crop failures and triggered famines on a global scale that killed more than 50 million people (Singh et al., 2018)."

Singh D, Seager R, Cook B I, Cane M, Ting M, Cook E and Davis M. Climate and the Global Famine of 1876-78[J]. 2018, Journal of Climate: 31(23): 9445-9467.

**2. Line 60. "a hundred-year agricultural failure" should be "a once-in-a-century"..**

A: Thank you, changed as suggested. The above-mentioned mistake has been corrected as suggested.

**3. Line 61-62. Why "the 1759 drought event did not cause serious social unrest" is a topic to be answered in the paper. So, the statement of "It can be seen that the societal adaptability at that time resolved the negative impact of the drought-driven agricultural failure" is not suitable be used here.**

A: Suggestion taken. We have changed this inappropriate statement to, "It can be inferred that there must have been some reasons that have offset the negative impact of this drought."

**4. Line 70. At present, Qinghai Province generally belongs to the Tibet. According to Figure 1, only part of Qinghai is included in the Northwest China.**

A: Suggestion taken. We have changed it to, "At present, Northwest China includes Shaanxi Province, Ningxia Hui Autonomous Region, Gansu Province, Xinjiang Uygur Autonomous Region, and the area east of Qinghai Lake in Qinghai Province."

**5. Line 71. It should be rainfed "crops can be planted in the eastern part of Northwest China". In the oases in Northwest China, if there is irrigation, crops can also be planted.**

A: Suggestion taken. We have changed it to, "rainfed crops can be planted in the eastern part of Northwest China."

**6. Figure1 a and b is suggested to combine into a single one.**

A: Suggestion taken. We have redrawn Figure1 a and b and merged them into Figure 1. The NAR and NWAR previously shown respectively on Figure 1a, and b are both presented now on Figure 1, using different color outlines to indicate different areas. The redrawn figure is shown as follows.

[Figure]

**Figure 1. The area inside the red border is NAR and the two provinces of NWAR (the area covered by the white oblique lines) during the Qing Dynasty. The provincial boundaries shown are those from the Qing Dynasty, drawn according to the Concise Historical Atlas of China (Tan, 1991).**

*7. Line 311. "drought, (1759)" is "drought (1759)".*

A: Thank you, changed as suggested. The above-mentioned mistake has been corrected as suggested.

*8. Line 369. "By 1759, the territory of the Qing Dynasty expanded further to Kashgar,……", does it mean "the Kashgar ,…… had under been controlled by central goverment of the Qing Dynasty".*

A: Thank you for your comments. We have changed it to, "By 1759, as soldiers of the Qing Dynasty continued to march westward, Hotan, Kashgar and the eastern part of the Pamirs, for example the Badakhshan (Badakhshan: now the Badakhshan Province in eastern Tajikistan and Wakhan Corridor in northeastern Afghanistan), had been controlled by the central government of the Qing Dynasty."

---

## Author Comment (AC2)

Dear anonymous referee,

Thank you very much for your valuable comments and constructive suggestions. Having carefully reviewed your feedback, we have gained significant insights and benefited greatly. We believe that revising according to your suggestions will greatly enhance the quality of our study. For instance, following your advice, we thoroughly polished the abstract again, adding points such as the significance of drought research in our study area and the feasibility of reconstructing drought processes and severity using historical literature. We carefully considered and corrected sentences with unclear semantics and insufficient emphasis, as mentioned in your feedback. Additionally, we supplemented and corrected the maps in the manuscript as per your suggestions, such as adding a larger background map and modifying the Digital Elevation Model (DEM) color gradient. Moreover, we corrected other inappropriate aspects pointed out by you in the manuscript. It is an honor for us to receive your review and guidance. The following are detailed point-to-point modifications made, with all changes already incorporated into the original manuscript. Once again thank you for your review and feedback!

**Anonymous referee, 18 Jan 2024**
*The authors Hao et al., present a reconstruction of the 1759 drought and the social response to such an extreme event in Norteast China. The article is well-structured, showing that the authors have dedicated a lot of time to documenting this event and its impacts on society. I missed, however, an analysis, or at least an hypothesis of the potential causes of such drought, I am not suggesting the use of climate models for this. However, I would have liked to see how their results are reconciled with those proposed in the Asian Monsoon Drought Atlas (Cook et al., 2010), which is supposed to be a reference in the region for reconstructing hydroclimate with annual resolution, and where it is shown that the period analyzed in that region is rather wet.*

Upon careful consideration, we have identified several factors that may account for the discrepancies between our research results and Cook's *Asian Monsoon Drought Atlas*:

1) Our study focuses on droughts which affected crops, and the growth period of crops is not year-round but primarily spans from late spring to early autumn, that is from May to July in Northwest China. Therefore, drought impacts on crops may not fully reflect the annual precipitation in this region.

2) The rugged terrain and significant elevation variations in Northwest China result in agricultural areas being predominantly situated in low-lying river valleys. This means, the agricultural areas are at different elevations from the forested areas where Cook's study utilized tree-ring proxy data. Drought conditions at lower elevations could lead to increased temperatures, facilitating upward air movement and convective precipitation in high-altitude mountainous forested areas, thereby maintaining relative humidity in the forested areas. Conversely, more moisture may precipitate in windward slope forests, potentially leading to water scarcity on leeward slopes and making low-lying farmland more susceptible to drought.

In summary, our study is based on reconstructions of drought beginning times and durations using historical documents, which primarily document the impacts of climate on human agricultural production, rather than the direct observations of natural vegetation growth, such as forests. Thus, differences in the temporal and spatial aspects of human-dependent crops and naturally growing vegetation may contribute to the deviations between our study and the *Asian Monsoon Drought Atlas*. Following your suggestions, we incorporated the above discussion into our manuscript and provided supporting references.

*1.The abstract needs a lot of polishing. First line doesnt make sense.. you need to write one or two lines giving some context on why this study is important for the region, etc. Then, you state that you were able to reconstruct droughts using historical documents, etc.*

A: Comment accepted. We have added sentences regarding the significance of the study for this area and the use of historical literature in studying drought.

*2.Line 10- Remove "of".*

A: Comment accepted. The above-mentioned mistake has been corrected as suggested.

*3.Line 15. 1759 is not needed here.*

A: Comment accepted. The above-mentioned mistake has been corrected as suggested.

*4.Lines 19-20 are confusing... You mean "with WORST?" Otherwise i dont understand the sentence.*

A: Thank you for pointing out the ambiguity. The original statement "*With better climatic conditions in the 18th century, China had higher agricultural harvest levels in the 18th century than in the 19th century.*" meant that the climatic conditions in the 18th century were better compared to the 19th century, resulting in overall higher agricultural yields in 18th-century China. Based on this, in the year 1759 of the 18th century, benefiting from good crop yields in regions outside of Northwest China, the government mitigated the impact of drought in the Northwest region on society through interregional grain allocation.

*5.Line 23. You need a couple of references there to support such a strong argument.*

A: Comment accepted. We have added a reference to support the viewpoint expressed in this sentence.

Haug, G., Gunther D., Peterson L., Sigman D., Hughen K. and Aeschlimann B.: Climate and the collapse of Maya civilization, Science, 299, 1731–1735, http://doi.org/10.1126/science.1080444, 2003.

*6.Line 25. I did not find Sapir, 2017 in the references.. is it Guha-Sapir?*

A: Thank you for pointing out this error. We have corrected the citation.

*7.FIgure 2. The maps could be improved by a lot. First, even though China is a big country, it would be helpful to include a location context map, within China. Then the DEM legen chosen is not informative at all (from 0 to 4000 m. asl), the colors are pretty similar. Either you want to*

*show that there are huge relief differences, or if not, maybe is better not to include the DEM at all. You already have an ok DEM in Figure 1.    I would consider removing it from Figure 2.*

A: Comment accepted. We have changed Figure 1 to include a larger map, illustrating the location of our study area within a broader region. Additionally, we have removed DEM in Figure 2.

*8.Figure 5. Cities, not Citys.*

A: Comment accepted. The above-mentioned mistake has been corrected as suggested.

*9.Figure 6. If its the same location as in Figure 2, why doing a different DEM? consider removing it.*

A: Thank you for pointing out this confusion. We have removed the DEM from Figure 6.

*10.The manuscript is already too long, but I honestly believed it would have more impact and would be more useful for the scientific community if 1) you try to reconcile your findings with those reported in the MADA, and 2) you try to give a potential cause for having such a drought.*

Thank you for highlighting this confusion. Our study relies on meteorological drought records extracted from historical literature, along with assessments of drought severity inferred from crop failures and relief efforts in affected areas. Therefore, the differences between our study results and Cook's *Asian Monsoon Drought Atlas* may be attributed to the rugged terrain of Northwest China, which resulted in varying elevations between agricultural and forested regions, leading to differences in drought and flood conditions. Additionally, the short growing season of crops may contribute to the inconsistency in climate conditions between agricultural areas, represented by crops, and forested areas, represented by tree rings.

In summary, our research is grounded in historical records, which offer direct accounts of climate impacts on human activities, aligning with the ultimate goal of climate change research: to save lives and property, and achieve sustainable societal development. Historical climate records primarily focus on regions with intense human activity and significant human disturbances, complementing studies using proxy data from natural sources like tree rings, stalagmites, and ice cores, which are less impacted by human interference.